# LSOS: An FG Position Method Based on Group Phase Ranging Ambiguity Estimation of BeiDou Pseudolite

**Heng Zhang [1,2,3] and Shuguo Pan [1,*]**

1    School of Instrument Science and Engineering, Southeast University, Nanjing 210096, China; zhh_seu@seu.edu.cn
2    State Key Laboratory of Satellite Navigation System and Equipment Technology, Shijiazhuang 050081, China
3    The 54th Research Institute of China Electronics Technology Group Corporation, Shijiazhuang 050081, China
*    Correspondence: psg@seu.edu.cn; Tel.: +86-137-7660-4834

**Abstract:** Due to the influence of indoor space environments, the carrier phase information obtained by the BeiDou pseudo-satellite often has the issue of cycle slips, which makes the user unable to carry out high-precision positioning. Aiming at the problem of ambiguity resolution (AR) and location in large-scale occluded space (LSOS), a factor graph (FG) position method based on group phase ranging ambiguity estimation of BeiDou pseudolite is proposed. Firstly, by introducing the principle of group phase period quantization and utilizing the multi-frequency characteristic of the BeiDou pseudo-satellite, the carrier phase propagation ambiguity of the BeiDou pseudo-satellite can be estimated quickly. On this basis, by introducing the shuffled frog leading algorithm (SFLA) assisted factor graph optimization location estimation method, the BeiDou pseudo-satellite positioning process in LSOS is realized. The experimental results show that the proposed method can solve the problem of fast estimation of ranging ambiguity of BeiDou pseudolite in LSOS, and the ranging accuracy can be improved to two wavelength ranges. In the further location experiment, it is found that the algorithm can not only guarantee the real-time location output but also improve the location precision to sub-meter level under the multi-frequency combination; the optimal location test precision is 9 cm, the maximum positioning error is 50 cm. This method successfully solves the problem wherein the BeiDou pseudo-satellite cannot provide real-time, continuous, and high-precision positioning in LSOS.

**Keywords:** BeiDou pseudo-satellite; integer ambiguity; group phase ranging; Global Navigation Satellite System (GNSS) rejection environment; positioning; large-scale space

## 1. Introduction

According to the National Human Activity Pattern Survey (NHAPS), people spend 86.9% of their time in indoor spaces. With the acceleration of urbanization, location-based service demand and applications have been expanding from outdoors to indoors. Many indoor scenes such as shopping malls, factories, parking and so on have a strong demand for location. Especially with the development of the new generation of artificial intelligence technology, a large number of unmanned equipment are expected to achieve continuous high-precision location service in indoor environment. However, due to the complex physical space, it is difficult to provide mature indoor positioning technology. The mainstream location methods such as WIFI, Bluetooth, near-ultrasonic and ultra-wideband (UWB) still need further research [1]. The most important problem currently is how to integrate with the existing user platform.

The biggest advantage of pseudolite indoor positioning technology is its good compatibility with GNSS [2–5]. Using the existing terminal can realize the data parsing without any hardware modification. In contrast, the research of pseudo-satellite indoor positioning technology is easier to achieve the promotion of landing. The research of this paper

mainly focuses on the indoor application of BeiDou pseudo-satellite technology. For ease of analysis, this article divides indoor spaces into typical large-scale space (large stadium, airport, station, exhibition hall, etc.), small-scale space (office, toilet, conference room, etc.), and narrow-long space (corridor, tunnel, underground pipe corridor, etc.). In this paper, the BeiDou pseudo-satellite positioning problem in large-scale occluded space is studied, and the application problems in other scenarios will be given in the following papers.

As we all know, carrier phase measurement is necessary for high precision positioning of GNSS. The GNSS navigation receiver will output the carrier phase as the original observation for the user. The specific form of carrier phase $\Phi_m$ is shown below.

$$\begin{cases} \Phi_1 = \|\mathbf{r}_t^1 - \mathbf{r}_u\| + c(\delta t_u - \delta T^1) - \lambda N^1 + \xi^1 \\ \Phi_2 = \|\mathbf{r}_t^2 - \mathbf{r}_u\| + c(\delta t_u - \delta T^2) - \lambda N^2 + \xi^2 \\ \quad\quad\cdots\cdots \\ \Phi_m = \|\mathbf{r}_t^m - \mathbf{r}_u\| + c(\delta t_u - \delta T^m) - \lambda N^m + \xi^m \end{cases} \tag{1}$$

In the formula, $\lambda$ is the wavelength of the received navigation signal, $\mathbf{r}_t^m$ is the coordinate position of the m pseudolite signal, $\mathbf{r}_u$ is the position of the user's receiver, $c$ is the speed of light (m/s), $\delta t_u$ is the clock difference of the user's receiver, $\delta T$ is the clock error of pseudolite signal, $N^m$ is the integer ambiguity of carrier phase estimation relative to k-channel pseudolite signal, and $\xi^m$ is the noise error..

As above, we obtain the range equations of the pseudo-satellite signals based on the carrier phase. Since the pseudo-satellite signals are generated by the same phase locked loop (PLL) control at each frequency during the design process, the clock difference of all signals is consistent. It can be found that the pseudo-satellite clock error $\delta t$ and the receiver clock error $\delta T$ can be eliminated by making the difference between the signals. Here we only give the difference equation based on the channel 1 signal. From this we can get the difference formula as follows:

$$\begin{cases} \Phi_2 - \Phi_1 = \|\mathbf{r}_t^2 - \mathbf{r}_u\| - \|\mathbf{r}_t^1 - \mathbf{r}_u\| + \lambda N^{21} + \xi^{21} \\ \Phi_3 - \Phi_1 = \|\mathbf{r}_t^3 - \mathbf{r}_u\| - \|\mathbf{r}_t^1 - \mathbf{r}_u\| + \lambda N^{31} + \xi^{31} \\ \quad\quad\cdots\cdots \\ \Phi_m - \Phi_1 = \|\mathbf{r}_t^m - \mathbf{r}_u\| - \|\mathbf{r}_t^1 - \mathbf{r}_u\| + \lambda N^{m1} + \xi^{m1} \end{cases} \tag{2}$$

From the above equation, it can be found that the distance difference between channels becomes a function related only to the true distance and the integer ambiguity of the signal. The formula contains M − 1 single-difference integer ambiguity and 3D user coordinate vector. M + 2 unknowns are actually larger than the number of observed equations. If we want to get the current position, we must complete the whole-cycle estimation process [6,7]. This process mainly includes initial position setting, search space selection, and ambiguity search strategy.

However, there is relatively little research on the issue of fast carrier phase ambiguity estimation of pseudolite in indoor space. Here, we refer to the scientific research results published by domestic and foreign experts and scholars in the past decade.

The Wuhan University team of Dr. Lee has carried out some deep research on the problem of rapid ambiguity fixation in indoor space. Dr. Lee's team developed a double-difference ambiguity fixation model and verified its performance by setting up a test environment in the laboratory. Furthermore, Li proposed a new indoor pseudolite positioning method using the combination of a robust unscented Kalman filter (RUKF) and partial AR (PAR) to improve the reliability of the position [8,9]. Liu and Yao propose a projected cancellation (PC) technique. This method can effectively improve the influence of nonlinear error by pseudo-range single difference according to the relationship between the base station and the user's virtual site. This method effectively improves the convergence speed of the traditional algorithm [10]. Dr. Wang's team proposes a technique for constructing an autonomous coordinate system without prior reference points. Through the fuzzy estimation of a three-step position estimation process, this method achieves the high precision

positioning of the transmitter station in centimeter-level [11]. Using the existing four GNSS antenna platforms, Dr. Zhao proposed a baseline-constrained ambiguity function method. This method can effectively improve the unreliable or inaccurate estimation solution of RTK localization, and further directly assist the estimation of altitude direction [12–15]. Professor Takuji Ebinuma of the Japanese Waseda University has been working on indoor pseudolite for many years and has extensive experience in rapid RTK positioning. In order to realize centimeter-level high-precision positioning in indoor environments, Takuji proposed a doppler RTK attitude estimation method based on the relative motion of two antennas. The position and attitude of the user terminal can be calculated by means of doppler shifts and phase measurement between the receiver antennas and transmitters [16–18]. Dr. Gan's team of Zhejiang University of Science and Technology proposed a carrier ambiguity resolution method based on the combination of pseudolite and UWB [19]. By using the high-precision ranging characteristic of UWB, the initial ambiguity of pseudolite is fixed. At present, research concerning integer ambiguity estimation is mainly based on the GNSS spatial signal. It is still less in the field of indoor location. Both Wuhan University and Seoul University have proposed a method to solve integer ambiguity quickly by using indoor RTK, which has achieved good positioning results in the experimental environment. Japanese Waseda University have developed a doppler-assisted fast ambiguity fixation method that provides cm-level positioning accuracy in indoor scenes. But according to the test data in the paper, the test is mainly carried out in slow motion, and the data is relatively stable. After the carrier phase cycle slips appear, the proposed method can't complete the fast ambiguity estimation. Not conducive to the protection of positioning continuity. In previous tests, the team's Dr. Gan proposed a location method based on Doppler's difference [20,21], but it is difficult to apply in large-scale space due to the constraint of initial location points. It is also difficult to popularize the pseudo-satellite positioning technology due to the heavy workload in the large-scale space. In the same way, the fingerprint localization [22] method proposed by Dr. Huang faces a huge workload when applied in large-scale space, which is not conducive to the popularization of pseudo-satellite positioning technology.

Although several typical schemes have been applied to the study of the carrier phase ambiguity estimation in indoor environment, there are still some problems in the comprehensive analysis. The main problems are as follows:

(1) The ambiguity estimation in the difference mode is based on the short baseline and is affected by the multipath signals of indoor reflectors. In general, the effectiveness of the difference information is limited to the local finite view-through space. It is not suitable for complex space application with indoor grid.

(2) In indoor spaces, the direct propagation distance of pseudo-satellite signal is limited. Affected by the range error of receiver from 3 to 10 m, the user can not realize the initial position estimation by the least square algorithm with the pseudo-range information obtained directly.

(3) The carrier phase space ambiguity search process of RTK is based on the continuous epoch without cycle slip, which usually needs to consider the setting of the search space. Assuming that there are L integers in the whole-cycle search space for each set of observations, $(m-1)^L$ times are required for the whole-cycle search space. The large amount of computation and the long estimation time make it difficult for the user to fix the ambiguity quickly when the indoor continuous grid space is switched.

(4) Due to the complexity of indoor physical space environments, shielding of moving bodies and strong reflection effect of adjacent reflectors will lead to frequent cycle slip phenomenon of the user receiver. The ambiguity of the integer needs to be re-estimated for each cycle slip, which is not conducive to the fast deblurring of measurement. Thus, it can be seen that it is difficult to provide good ranging accuracy for indoor location estimation based on the current carrier phase ambiguity estimation method.

Therefore, the question concerning how to solve the high-precision ranging problem is the key to realize the absolute positioning in LSOS. In response to this problem, the main

work of this article is as follows: In the second chapter, as the key of BeiDou pseudo-satellite phase feature extraction and group phase feature ranging application, the basic theory of BeiDou pseudo-satellite network platform is introduced. In the third chapter, according to the multi-frequency characteristics of BeiDou pseudolite, the phase quantization period theory and the phase whole cycle estimation process are described in detail. In the fourth chapter, the performance of the proposed method is analyzed in detail from two aspects of measurement and simulation. Finally, on the basis of the existing results, we summarize the technical advantages of the current method and provide some follow-up research directions.

## 2. System Overview

BeiDou array pseudo-satellite (BDAPS) is an independent indoor positioning system which can realize indoor independent networking without relying on any external connection. According to the characteristics of indoor space environments, a centralized array network, Beacon Cascade Network, and space distributed network can be built in indoor viewing environment. The base station is designed with multi-elements of the same source. The time-frequency reference parameters of the multi-channel pseudo-satellite signals are generated by a common clock source, and the multi-channel pseudo-satellite signals have strictly consistent time-frequency characteristics. Thus, each pseudo-satellite generates the same satellite clock error information. The core of BDAPS consists of time-frequency control unit, baseband signal generating unit, radio frequency unit, and antenna unit. The block diagram of the base station composition is shown in Figure 1.

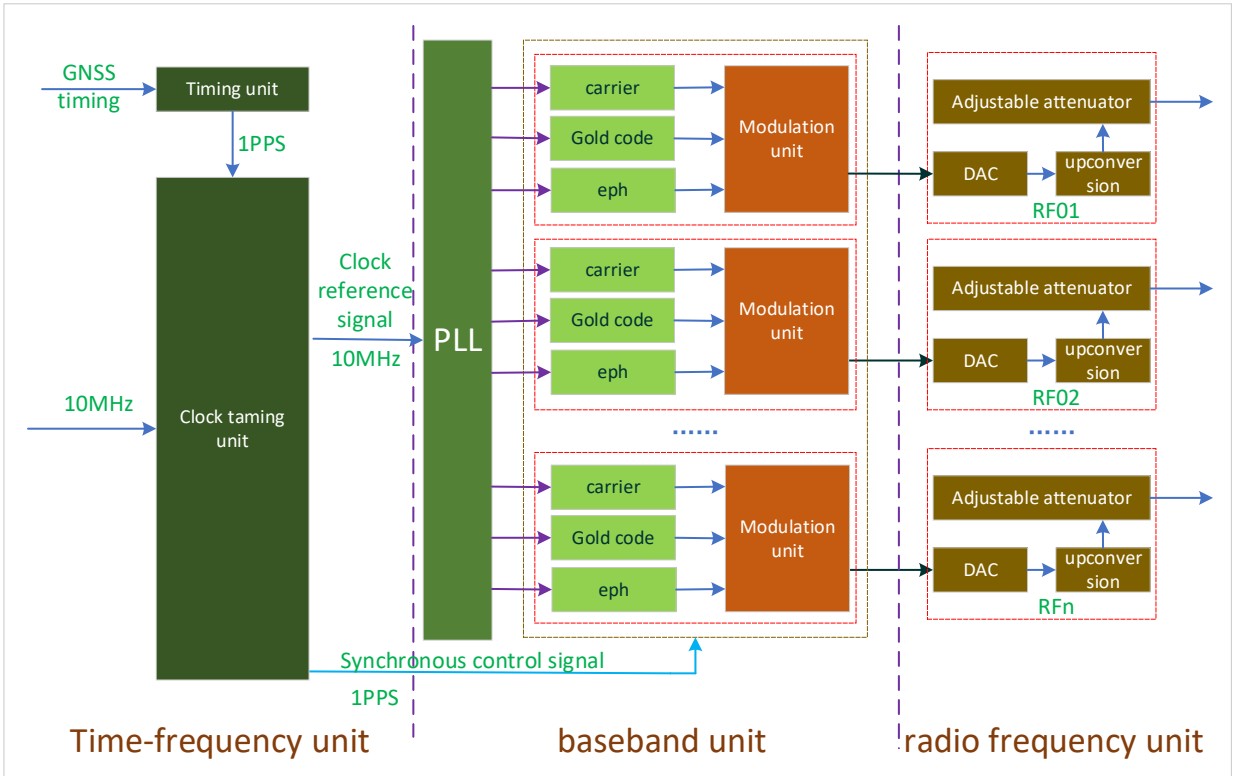

**Figure 1.** Principle of BDAPS signal generation.

## 3. Algorithmic Principle

The carrier phase generally consists of two parts: the integer part of the carrier and the fractional part of the carrier. The fractional portion of carrier phase can be estimated accurately through real-time frequency. Therefore, carrier group phase estimation focuses on how to use the fractional portion of carrier phase information to estimate the integer portion of the carrier phase. The pseudo-satellite in this paper has two technical advantages in the design process: (1) the homologous array pseudo-satellite is controlled by the same

clock source, and the signal has strict consistency in the process of clock deviation and channel delay; (2) in the design process of the BeiDou multi-frequency pseudo-satellite signal, the nominal frequency has strict consistency, which provides a basis for our research on the multi-frequency phase measurement method indoors. Based on this advantage, the theory of group phase ranging is introduced in this paper. The phase cumulative characteristic of pseudolite signal is estimated by using the period characteristic of multi-level phase deviation produced by multi-frequency signal during phase comparison. Thus, the accurate measurement of distance measurement can be completed without depending on the transmission time.

### 3.1. Phase Quantization Process

Generally, frequency and phase belong to the inherent properties of periodic signals. The change of frequency will inevitably cause the change of phase. One of the most remarkable characteristics of periodic signals is the periodic change of phase difference introduced by frequency difference. Usually, the change of frequency and phase can be converted into each other. The high-precision measurement of frequency can be converted into the measurement of the phase comparison of two periodic signals. The phase change can reflect the regular change between periodic signals more precisely.

Phase information is a physical quantity, which reflect the state of periodic signals at any moment. The process of phase change among several periodic signals directly reflects the process of quantization between periodic signals in time. The following takes any two frequency points as an example to introduce the phase change relationship between periodic signals, as shown in Figure 2.

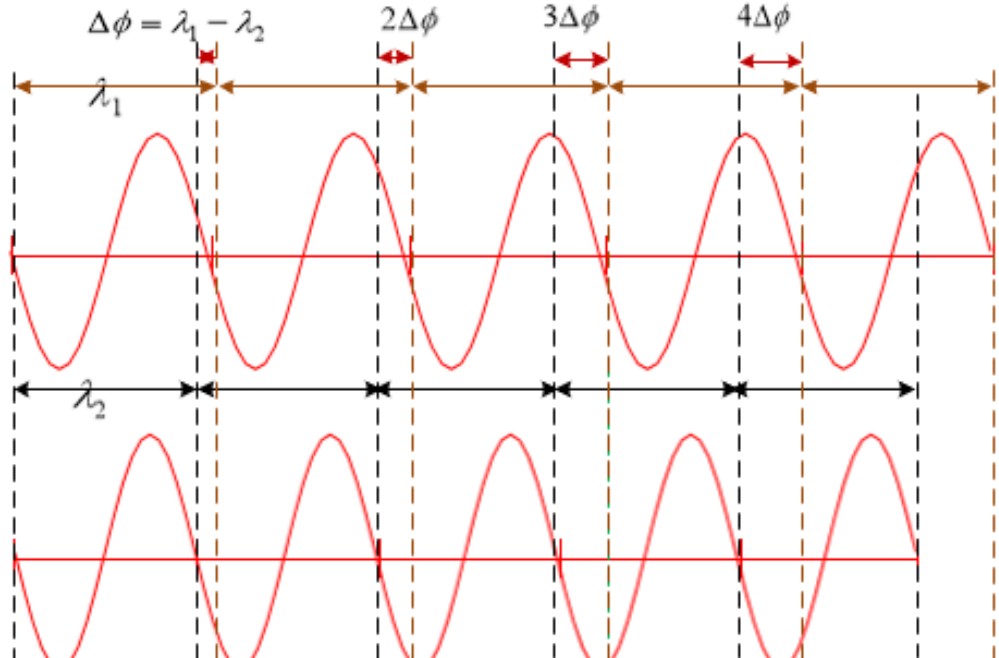

**Figure 2.** Phase quantization relationship between periodic signals.

Suppose that the selected signal frequencies are $f_1$ and $f_2$, $f_1 < f_2$. The corresponding signal period is $T_1$ and $T_2$. If the phase coincidence point between the two signals is identified as phase point 0, it can be seen from Figure 2 that if the phase of the two signals is to be the same, the time-domain relationship between the two frequency signals should be satisfied

$$t = \frac{M}{f_2 - f_1} \qquad f_0 = (f_1, f_2) \qquad T_{\min} = \frac{1}{(f_1, f_2)} \tag{3}$$

M is a positive integer, when M = 1, $(f_1, f_2)$ is the maximum common factor frequency of $f_1$ and $f_2$, and $T_{\min}$ is a least common multiple period between two signals, which reflects the variation of the phase difference of the periodic signal in the least common multiple period and the variation of the temporal and spatial signal. The quantization is generally a quantitative change relation with a unit interval of $f_2 - f_1$ as the basis.

In this case, the phase difference between the two phases can be expressed as

$$\Delta t = n \frac{f_0}{f_1 f_2} \tag{4}$$

n is a positive integer, $n = 0, 1, 2, 3, \ldots \ldots f_1/f_0$, $\frac{f_0}{f_1 f_2}$ is the quantized phase shift discrimination rate between the two signals, namely the unit quantum. The change of phase difference is always an integer multiple of the phase quantum, which can be from 0 to $f_1/f_0$.

From the above analysis, there is always a minimum quantization interval in the whole quantization process. The interval is the quantization resolution between two signals, which shows a quantization relation between the mutual phases. This change in phase difference is a step change in the smallest unit, which is called the quantized phase-shift resolution. The least common multiple of a periodic signal is shown in time as the least common multiple period between any two periodic signals. There is no such thing as the same phase difference in any given least common multiple cycle. The variation of phase difference is an integral multiple of phase shift resolution, which reflects the whole-cycle phase estimation during periodic phase coincidence.

### 3.2. Group Phase Estimation

According to frequency signal phase quantization processing in the previous section, the unit quantization theory can be better applied to phase measurement. The phase difference between periodic signals varies periodically in least common multiple, and any periodic phenomenon can be represented by a group. After further study of this phenomenon, it is found that the variation features have some symmetry, which is called group phase theory in the following. The group phase character belongs to the finite periodic cycle class. According to the different frequency relations, the characteristics of multi-frequency point group phase estimation are analyzed.

The set of phase differences in the least common multiple periods between any two periodic signals is called a cell group, and each cell group consists of several subsets. If we think of each least common multiple element in the actual least common multiple cycle as a subset, each subset can be used as $GU_1, GU_2, \ldots GUn$.

$$G_n = \{t | \Delta t^n = 1\} \tag{5}$$

$$G_n = \{GU_1 \quad GU_2 \quad \ldots \quad GU_n\} \tag{6}$$

Here t is the generator and n is the order. The phase difference group is a finite cyclic group, isomorphic to the addition group. This group can be discussed in terms of the theory of finite additive groups. When the nominal value between two periodic signals is the same, the generator in a minimum common multiple periods is not unique, but the minimum generator between signals is unique.

Obviously, the minimum generator is the highest phase shift resolution between two signals. In the phase comparison and frequency measurement, the most concerned is the quantized phase shift resolution. A cyclic group of order n has $\varphi$ (n) generators, and $\varphi$ (n) is Euler function. When n is an integer greater than 1, and n = $pk_n$ is the standard factorization of n. n has T(n) = $(k_1 + 1) (k_2 + 1) \ldots (k_m + 1)$ positive factors, where T(n) of n is the number of positive factors of n. Then cyclic groups of order n have only T(n) cyclic group subgroups. If the least common multiple period between two periodic signals is $T_{\text{minc}}$, if $f_1 < f_2$, then the order of the phase group is $T_{\text{minc}} \times f_2$. Obviously, such a cyclic phase group has a strong regularity, and it is this regularity that makes all kinds of comparisons work smoothly.

In any $GU_i$, the phase changes between signals are complex and changeable. In the same cyclic group, the changes are in line with the characteristics of cyclic group. The construction of cyclic groups is completely determined by the generator (that is, the resolution of the quantized phase shift between the two signals).

$$GP_n = \begin{Bmatrix} G_1 & G_2 & \dots & G_{Num} \end{Bmatrix} \tag{7}$$

Assuming that there are N frequency points of the signal, the order of the frequency is $f_1 \quad f_2 \quad \dots \quad f_N$. Therefore, the corresponding number of least common multiple periods can be obtained according to different frequency combinations, and the total number of phase quantization period combinations of $Num = C_N^2 + C_N^3 + C_N^4 + \dots + C_N^N$ species can be obtained. The group phase quantization period can be expressed as $T = \begin{bmatrix} T_1 & T_2 & \dots & T_{Num} \end{bmatrix}$. Thus, the element group can be extended to group-phase subgroup cells with different number of periods, and the characteristic period can be estimated from the minimum resolution element, the element group to the group-phase multi-level least common multiple.

$$GP_n = \begin{Bmatrix} G_1 & G_2 & \dots & G_{Num} \end{Bmatrix} \tag{8}$$

*3.3. Group Phase Ranging Estimation*

The process of group phase quantization is actually the refinement and segmentation process of signal spatial transmission. Through group phase quantization, the phase information of signal is accurately classified. The following ranging process is based on this expansion.

In order to better evaluate the application of the group phase period quantization principle in the pseudo-satellite system, the group phase period of the BeiDou pseudo-satellite under multi-frequency is analyzed in combination with the common frequency points of BeiDou. A total of six commonly used frequency points are selected in the following table, as shown in Table 1.

**Table 1.** List of existing frequency points of the BeiDou signal.

| Serial Number | Frequency | Value (MHz) |
|:---:|:---:|:---:|
| 1 | B1I | 1561.098 |
| 2 | B2I, B2b | 1207.14 |
| 3 | B3I | 1268.52 |
| 4 | B2a | 1176.45 |
| 5 | B1C | 1575.42 |

Combined with the maximum common factor frequency characteristics between frequencies, the first step of this paper is to analyze the maximum common factor frequency characteristics between frequencies of dual-frequency signals. Influenced by the reference frequency, the common maximum common factor frequency characteristics are shown in the combination estimation process of multiple frequencies. As shown in Table 2, four maximum common factor frequency values are obtained in total from the selected frequencies.

It can be seen that the group phase quantization period after multi-frequency combination divides the signal transmission process into four segments and converts the group phase quantization period into a constant representation of distance

$$\mathbf{S} = [146.5261, 29.3052, 20.9323 \ 14.6526] \tag{9}$$

Normally, the standard height of the indoor space is 2.8~3.3 m, the underground parking lot is 2.5 m, the stadium, station and exhibition center is 30 m, the tunnel is 5.8 m, and the underground pipe gallery is 4.6~6 m. BeiDou pseudolite has abundant group phase quantization period segmentation characteristics, which fully meets the requirements of indoor underground space ranging.

**Table 2.** The common maximum common factor frequency characteristics.

| 2.046 MHz | 10.23 MHz | 20.46 MHz | 14.322 MHz |
|---|---|---|---|
| (B1I, B2I) | (B1C, B2a) | (B3I, B2I) | (B1I, B1C) |
| (B1I, B3I) | (B2I, B2a) | (B1C, B2I) | - |
| (B1I, B2a) | (B3I, B2a) | (B1C, B3I) | - |
| (B1I, B2I, B2a) | (B1C, B2a, B3I) | - | - |
| (B1I, B2I, B3I) | (B1C, B2a, B2I) | - | - |
| (B1I, B2I, B1C) | (B2I, B3I, B2a) | - | - |
| (B1I, B3I, B2a) | (B1C, B2a, B3I, B2I) | - | - |
| (B1I, B3I, B1C) | - | - | - |
| (B1I, B1C, B2a) | - | - | - |
| (B1I, B2I, B1C, B2a) | - | - | - |
| (B1I, B2I, B3I, B2a) | - | - | - |
| (B1I, B2I, B3I, B1C) | - | - | - |
| (B1I B3I, B1C, B2a) | - | - | - |

The range estimation process in detail is given below. In order to facilitate analysis, the putative $f_1/f_2/f_3$ three frequency signals, and $f_1 < f_2 < f_3$. From this, we can get the quantization period set of group phase under different frequency combinations. Then the group phase quantization period set is

$$T = \begin{bmatrix} T_1 & T_2 & T_3 & T_4 \end{bmatrix} \tag{10}$$

Due to the influence of channel hardware and software delay, the pseudo-satellite signal of each channel is usually not 0. However, due to the strict time-frequency characteristics of pseudolite, it can be assumed that each launch channel has the same phase delay. In the phase comparison process, Formula (17) is the group phase quantization cycle set. $\Delta\phi_0$ is the minimum period element for realizing the same phase delay again in the frequency point combination. Assume that the initial phase difference of $\Delta\phi$, ranging error is introduced in the initial phase difference

$$\Delta\rho = \lambda \frac{\Delta\phi}{\Delta\phi_0} \tag{11}$$

Since the distance error introduced by phase difference is a constant value, it will not affect the variation of group phase quantization period in practical application. Therefore, the following distance estimation process is no longer considered alone and the default initial phase is 0.

First of all, the distance estimation process and the pseudo-satellite positioning process require at least four pseudo-satellite signals. It is assumed that there are n pseudo-satellites, the position coordinate array $\mathbf{R}_s$ and that the reference point coordinates of the spatial virtual grid based on the covering domain are

$$\mathbf{R}_s = \begin{bmatrix} x_s^1 & y_s^1 & z_s^1 \\ x_s^2 & y_s^2 & z_s^2 \\ \dots & \dots & \dots \\ x_s^n & y_s^n & z_s^n \end{bmatrix} \quad \mathbf{R}_b = \begin{bmatrix} x_b^1 & y_b^1 & z_b^1 \\ x_b^2 & y_b^2 & z_b^2 \\ \dots & \dots & \dots \\ x_b^m & y_b^m & z_b^m \end{bmatrix} \tag{12}$$

The measurement equation is constructed. In the original observation information of pseudo-satellite output by receiver, the pseudo-distance can be directly obtained from the pseudo-satellite signal, and there is no ambiguity distance information. Doppler information is related to the motion state of the receiver, there is no ranging estimation error. So, if the pseudorange expressed in $\rho_s$, doppler use $f_d$, said the distance of the current

time is estimated to be $\rho_s(n) = \rho_s(n-1) + \lambda f_d \Delta t$, thus it is concluded that the user receiver to the location of the estimating equations

$$\boldsymbol{\rho}_s(k) = \|\mathbf{R}_s - \mathbf{r}(k-1)\| + \Delta t - \Delta T_s + \varepsilon \tag{13}$$

In the formula, $\boldsymbol{\rho}_s(k) = \begin{bmatrix} \rho_s^1(k) \\ \rho_s^2(k) \\ \cdots \\ \rho_s^n(k) \end{bmatrix}$, k is a positive integer, $k \geq 1$. $\Delta t$ is receiver clock error.

$\Delta T_s$ is pseudo-satellite clock error. if $\mathbf{r}(n) \in \mathbf{R}_b$, then $\min(\boldsymbol{\rho}_s(k) - (\|\mathbf{R}_s - \mathbf{r}(k)\| + \Delta t - \Delta T_s + \varepsilon))$ is recorded as the selected optimal estimated position. Therefore, the transmission time of pseudolite signals in the current position can be deduced as $t = [t_1, t_2, \ldots t_n]$. According to the interval characteristic of each signal transmission time, the whole group phase period number $T_c = \begin{bmatrix} T_1 & T_2 & \ldots & T_n \end{bmatrix}$ of each transmission time is deduced.

The group phase segmentation process further solves the problem of quantifying the cyclic period of the group phase, transforming the problem into the problem of phase difference estimation within the least common multiple period. It is assumed that the inter-frequency carrier phase difference between the currently acquired multi-frequency signals is

$$\Delta\boldsymbol{\Phi} = [\Delta\Phi_1 \quad \Phi_2 \quad \ldots \quad \Delta\Phi_n] \tag{14}$$

According to the principle of minimum resolution of group phase above, the minimum resolution value of group phase under each path can be obtained as

$$\Delta\boldsymbol{\varphi} = [\Delta\varphi_1 \quad \Delta\varphi_2 \quad \ldots \quad \Delta\varphi_n] \tag{15}$$

From this, $\mathbf{N} = [N_1 \quad N_2 \quad \ldots \quad N_n]$, the formula for estimating the difference in the integer of the least common multiple is obtained

$$\min(\Delta\boldsymbol{\varphi} \cdot \mathbf{N} - \lfloor \Delta\boldsymbol{\varphi} \cdot \mathbf{N} \rfloor - \Delta\boldsymbol{\Phi}) \tag{16}$$

Thus, the distance value of each signal is

$$\widehat{\boldsymbol{\rho}} = c\mathbf{T}_c + \lambda(\mathbf{N} + \Delta\boldsymbol{\psi}) \tag{17}$$

In the formula, $\lambda$ is the signal wavelength of the lower frequency in the frequency difference. $\Delta\boldsymbol{\psi} = [\Delta\psi_1 \quad \Delta\psi_2 \quad \ldots \quad \Delta\psi_n]$, $\Delta\psi_n$ is the signal phase within a cycle with low frequency in the frequency difference at the current time.

*3.4. FG Optimization Positioning Method Assisted by SFLA*

3.4.1. BDAPS FG Model

The data fusion factor model of BDAPS is built on the basis of the existing original observation data. The system factor graph model is constructed by observing the navigation state in the time domain and all available navigation measurements. On this basis, two kinds of models are established, which are measurement model and process model.

The measurement model is a posteriori probability estimation model, which represents the conditional probability of the measured value obtained at time K under the $\mathbf{X}_k$ of the navigation state set at that time. In a factorial graph, each measurement corresponds to a factor node in the factorial graph. The ranging process is the primary process used to construct the positioning equation. It is assumed that the ranging value at k-time is $\rho_{r,t}^s$. When there is n-channel signal, the pseudo-range of n-channel signal is obtained

$$\rho_{r,k}^s = [\rho_{r,k}^1 \quad \rho_{r,k}^2 \quad \cdots \quad \rho_{r,k}^N] \tag{18}$$

Since the indoor environment is usually near-earth, the effect of ionospheric delay can be ignored. The influence of troposphere generally defaults that the tropospheric delay is

invariable under the same environment region. The measurement equation based on single difference of distance can be expressed as

$$
\begin{bmatrix} r^{21}_{r,k} \\ r^{31}_{r,k} \\ \cdots \\ r^{N1}_{r,k} \end{bmatrix} = \begin{bmatrix} \|\mathbf{p}_{r,k} - \mathbf{p}^2_k\| - \|\mathbf{p}_{r,k} - \mathbf{p}^1_k\| \\ \|\mathbf{p}_{r,k} - \mathbf{p}^3_k\| - \|\mathbf{p}_{r,k} - \mathbf{p}^1_k\| \\ \cdots \\ \|\mathbf{p}_{r,k} - \mathbf{p}^N_k\| - \|\mathbf{p}_{r,k} - \mathbf{p}^1_k\| \end{bmatrix} + \mathbf{w}^s_{r,k}
\tag{19}
$$

In the formula, $\mathbf{p}_{r,k}$ is the user's position at k-time, $\mathbf{p}_{r,k} = [x_k \quad y_k \quad z_k]$, and $\mathbf{p}^s_k$ is the space position of the s-type BeiDou pseudo-satellite at k-time. Therefore, the pseudo-range measurement information equation is a function related to the position of the pseudo-satellite and the position of the receiver. The position of the BeiDou pseudolite is usually demarcated by outdoor RTK and total station in advance. The position of the BeiDou pseudolite is known here. We can see that $h(\mathbf{p}_{r,k})$ is only a function of the user's position at K time. $\mathbf{w}^s_{r,k}$ is the noise error introduced by spatial noise, and we consider it a random Additive white Gaussian noise.

$$
h(\mathbf{p}_{r,k}) = \begin{bmatrix} \|\mathbf{p}_{r,k} - \mathbf{p}^2_k\| - \|\mathbf{p}_{r,k} - \mathbf{p}^1_k\| \\ \|\mathbf{p}_{r,k} - \mathbf{p}^3_k\| - \|\mathbf{p}_{r,k} - \mathbf{p}^1_k\| \\ \cdots \\ \|\mathbf{p}_{r,k} - \mathbf{p}^N_k\| - \|\mathbf{p}_{r,k} - \mathbf{p}^1_k\| \end{bmatrix} + \mathbf{w}^s_{r,k} \quad \mathbf{d}\boldsymbol{\rho}^s_{r,k} = \begin{bmatrix} r^{21}_{r,k} \\ r^{31}_{r,k} \\ \cdots \\ r^{N1}_{r,k} \end{bmatrix}
\tag{20}
$$

In equation, $\mathbf{d}\boldsymbol{\rho}^s_{r,k}$ is not obtained from the original observation information of the receiver, but estimated from the multi-frequency group phase characteristics. Here, we assume that there is a distance estimator obtained from the combination between group m frequencies, then the distance value of the current path signal can be expressed as

$$
g(ps^s_k) = hi(p^1_k, p^2_k, \ldots, p^m_k)
\tag{21}
$$

The estimation process of distance measurement value at the current time also conforms to the measurement model, thus the corresponding factor node $f^{meas}_{p,k}(ps^s_k)$ can be expressed as an error function, such as formula

$$
f^{meas}_{p,k}(ps^s_k) = d(err_k(ps^s_k, g(ps^s_k)))
\tag{22}
$$

Furthermore, the process also conforms to the white Gaussian noise process, and the error function is proportional to a probability model

$$
\begin{aligned}
P(z^{meas}_{p,k} \big| ps_k) &\propto f^{meas}_{p,k}(ps^s_k) \\
&= \exp(-\tfrac{1}{2}\|err_k(ps_k, z^{meas}_{p,k})\|^2_{Y_k}) \\
&= \exp(-\tfrac{1}{2}\|ps_k - z^{meas}_{p,k}\|^2_{Y_k})
\end{aligned}
\tag{23}
$$

The factor node $f^{meas}_k(\mathbf{p}_{r,k})$ at the current moment K is set as the error function at the current moment, where $d(\cdot)$ represents the cost function corresponding to the navigation state $\mathbf{p}_{r,k}$ and the measured value $\mathbf{d}\boldsymbol{\rho}^s_{r,k}$ at the current moment.

$$
f^{meas}_k(\mathbf{p}_{r,k}) = d(err_k(\mathbf{d}\boldsymbol{\rho}^s_{r,k}, h(\mathbf{p}_{r,k})))
\tag{24}
$$

so the error function process of the above equation can be converted into a probabilistic process to solve the following equation.

$$
\begin{aligned}
P(\mathbf{d}\boldsymbol{\rho}^s_{r,k} \big| \mathbf{p}_{r,k}) &\propto f_k(\mathbf{p}_{r,k}) \\
&= \exp(-\tfrac{1}{2}\|err_k(\mathbf{d}\boldsymbol{\rho}^s_{r,k}, h(\mathbf{p}_{r,k}))\|^2_{\Gamma_k}) \\
&= \exp(-\tfrac{1}{2}\|\mathbf{d}\boldsymbol{\rho}^s_{r,k} - h(\mathbf{p}_{r,k})\|^2_{\Gamma_k})
\end{aligned}
\tag{25}
$$

$\Gamma_k$ is the measurement noise covariance at k-time, and $h(\mathbf{p}_{r,k})$ is the nonlinear measurement function of the measurement model at k-time. The square mahalanobis distance in the formula calculates the residuals of the single difference between the measured $h(\mathbf{p}_{r,k})$ and the distance estimated by the group phase at the current K time.

From the cost function of the factor node, it can be seen that the factor node $f_k^{meas}(\mathbf{p}_{r,k})$ is only related to the navigation state quantity $\mathbf{p}_{r,k}$ at the current moment.

The process model is a prior estimation model, which represents the process of estimating the state of the system from the previous time to the current time. In the application process of BDAPS, there are mainly two stages involved in this process: (1) the estimation process of ranging measurement based on doppler; (2) Position estimation process based on doppler. The following two process factor graph models are modeled respectively.

In the estimation of ranging measurements based on doppler, the pseudo-satellite signal doppler feature is caused by the receiver movement. Based on the doppler measurement $\mathbf{d}_{r,k}^s = (d_{r,k}^1, d_{r,k}^2, \ldots, d_{r,k}^N)$ of K at the current moment, the current doppler-based velocity relation.

$$\mathbf{y}_{r,k}^d = (\lambda d_{r,k}^1, \lambda d_{r,k}^2, \ldots, \lambda d_{r,k}^N)^T \tag{26}$$

The current time estimation is obtained by smoothing the estimated range and doppler values or the carrier phase difference between the fore-and-aft epoch. After the predicted value of the system state is obtained, the residual value is calculated according to the measured value of the current time obtained in the process of group phase estimation.

$$f_{p,k}^{pro}(\mathbf{ps}_k) = d(err_k(\mathbf{ps}_k, g(\mathbf{ps}_k))) \tag{27}$$

Similarly, the probability estimation process is transformed from the above equation into

$$
\begin{aligned}
P(\mathbf{ps}_k | \mathbf{ps}_{k-1}) &\propto f_k^{pro}(\mathbf{ps}_k) \\
&= \exp(-\tfrac{1}{2}\|err_k(\mathbf{ps}_k, g(\mathbf{ps}_k))\|_{\Gamma_k}^2) \\
&= \exp(-\tfrac{1}{2}\|\mathbf{ps}_k - g(\mathbf{ps}_k)\|_{\Gamma_k}^2)
\end{aligned}
\tag{28}
$$

In the process of location estimation based on doppler, the system status $\mathbf{p}_{r,k-1}$ at the last moment and the system status transition function are used to predict the system status value $\hat{\mathbf{p}}_{r,k}$ at the current K moment. Affected by the receiver's doppler drift, we usually get the doppler value from the speed doppler $dop_{v,k}^N$ and the doppler drift doppler $dop_{cdf,k}^N$, the correlation equation can be expressed as

$$d_{r,k}^N = dop_{v,k}^N + dop_{cdf,k}^N \tag{29}$$

$$
\begin{aligned}
\mathbf{y}_{r,k}^d &= \mathbf{dop}_{v,k} + \mathbf{dop}_{cdf,k} \\
&= \lambda \left( \begin{bmatrix} dop_{v,k}^1 \\ dop_{v,k}^2 \\ \ldots \\ dop_{v,k}^N \end{bmatrix} + \begin{bmatrix} dop_{cdf,k}^1 \\ dop_{cdf,k}^2 \\ \ldots \\ dop_{cdf,k}^N \end{bmatrix} \right)
\end{aligned}
\tag{30}
$$

Therefore, the relation equation between velocity and position information can be written as

$$\mathbf{y}_{r,k}^d = \mathbf{v}_{r,k}^T \mathbf{e}_{r,k}^s + \varepsilon_{r,k}^d \tag{31}$$

$\mathbf{v}_{r,k}$ is the speed value of the user, $\Delta t$ is the unit time interval of the data output, the interval time is not necessarily 1 s, under normal circumstances, the data rate of the user terminal output has 1 Hz, 5 Hz, 10 Hz, 20 Hz and so on. $\mathbf{e}_{r,k}^s$ is the direction vector between

the current position and the transmitting antenna position of BDAPS at the estimated position at the current K time.

$$\mathbf{v}_{r,k} = \frac{\mathbf{p}_{r,k} - \mathbf{p}_{r,k-1}}{\Delta t} = \begin{bmatrix} \frac{x_k - x_{k-1}}{\Delta t} \\ \frac{y_k - y_{k-1}}{\Delta t} \\ \frac{z_k - z_{k-1}}{\Delta t} \end{bmatrix} \qquad \begin{aligned} \mathbf{e}_{r,k}^s &= \begin{bmatrix} \mathbf{e}_{r,k}^1 & \mathbf{e}_{r,k}^2 & \cdots & \mathbf{e}_{r,k}^N \end{bmatrix} \\ &= \begin{bmatrix} e_{r,k,x}^1 & e_{r,k,x}^2 & \cdots & e_{r,k,x}^N \\ e_{r,k,y}^1 & e_{r,k,y}^2 & \cdots & e_{r,k,y}^N \\ e_{r,k,z}^1 & e_{r,k,z}^2 & \cdots & e_{r,k,z}^N \end{bmatrix} \end{aligned} \tag{32}$$

Thus, the doppler velocity parameter is obtained as an equation relating the receiver's current time position to the previous time position, the pseudolite position and the forward and backward time interval units. $\mathbf{p}_k^N \; \mathbf{p}_{r,k-1} \; \Delta t$ are known quantities at the current moment. So the equation of observation is $gg(\mathbf{p}_{r,k}) = \mathbf{v}_{r,k}^T \mathbf{e}_{r,k}^s$. Through the above equation, the doppler velocity error function can be expressed as

$$f_{v,k}^{pro}(\mathbf{p}_{r,k}) = d(err_k(\mathbf{y}_{r,k}^d, gg(\mathbf{p}_{r,k}))) \tag{33}$$

the corresponding error function is expressed by the equivalent probability estimation process as

$$\begin{aligned} P(\mathbf{p}_k \big| \mathbf{p}_{k-1}, z_k^{pro}) &\propto f_{v,k}^{pro}(\mathbf{p}_{r,k}) \\ &= \exp(-\tfrac{1}{2} \| err_k(\mathbf{p}_{r,k}, z_k^{pro}) \|_{\Lambda_k}^2) \\ &= \exp(-\tfrac{1}{2} \| \mathbf{p}_{r,k} - z_k^{pro} \|_{\Lambda_k}^2) \end{aligned} \tag{34}$$

### 3.4.2. BDAPS Factor Graph Framework

In the above process, we set up the corresponding measurement model and process model for the observation domain and the location domain respectively. It can be found that the factor nodes of the observation domain and the location domain are associated with the current time variable node and the previous time variable node. The state variables of BDAPS system are parameters concerning distance measurements and user position coordinates. Let the combined state quantity be as follows:

$$A_k = [\rho s_1, \rho s_2, \ldots, \rho s_N, x_k, y_k, z_k] \tag{35}$$

According to the construction process in the previous section, all of the measurement information obtained by farmers in the navigation system is integrated to build the measurement set $Z_k$. The Maximum a posteriori (MAP) estimation density estimation is used to find the maximum a posteriori value of the navigation state set $A_k$ for information fusion. Then the cooperative navigation problem is transformed into an equivalent nonlinear optimization problem.

$$\begin{aligned} \widetilde{A}_k &= \underset{A_k}{\arg\min}[-\ln P(A_k|Z_k)] \\ &= \underset{A_k}{\arg\min}[-\ln(\prod_{i=u}^{k} P(x_i \big| x_{i-1}, \kappa_i^{pro}) \prod_{i=u}^{k} P(\kappa_i^{meas} \big| x_i))] \\ &= \underset{A_k}{\arg\min}[\sum_{i=u}^{u} \| f_{all,i}^{pro}(x_{i-1}, \kappa_i^{pro}) - x_i \|_{H_i}^2 + \sum_{i=u}^{u} \| f_{all,i}^{meas}(x_i) - \kappa_i^{meas} \|_{I_i}^2 ] \end{aligned} \tag{36}$$

$I_i$ is the measurement noise covariance, $H_i$ is the process noise covariance, $f_{all,i}^{meas}(x_i)$ is the nonlinear measurement function of the measurement model, $f_{all,i}^{pro}(x_{i-1}, \kappa_i^{pro})$ is the nonlinear state transfer function of the process model, $\| \|^2$ is the square Markov distance. The estimation process of the state vector becomes the problem of finding the minimum point of the nonlinear function. The factorial optimization model architecture for BDAPS is shown in Figure 3.

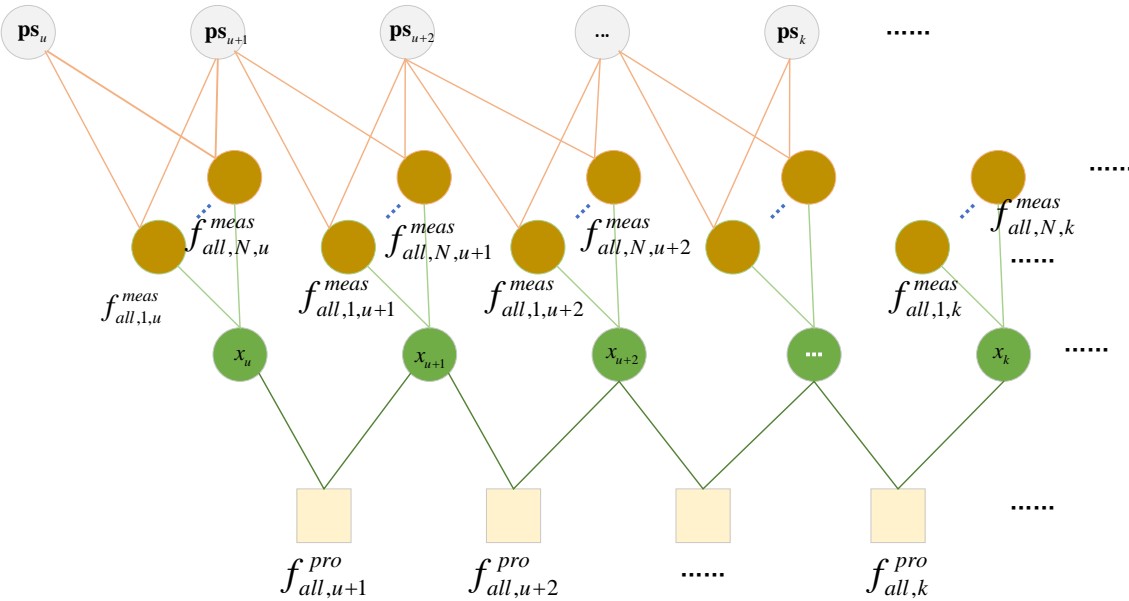

**Figure 3.** The factorial optimization model architecture for BDAPS.

### 3.4.3. Optimal Estimation Method of Factor Graph Based on SFLA

To solve the current nonlinear problems, the least squares method is usually used to make the optimal estimation by Taylor expansion of the covariance function and the derivation of the equation. However, due to the limitation of propagation distance and spatial noise, the conventional first-order Taylor expansion neglects the effect of the expansion on the optimal position estimation. In the early test process, it is difficult to get an effective solution in many cases. As one of the current intelligent group search algorithms, the SFLA algorithm can make up for the shortcomings in the optimal solution estimation of nonlinear functions. The algorithm processing block diagram of this paper is given in Figure 4.

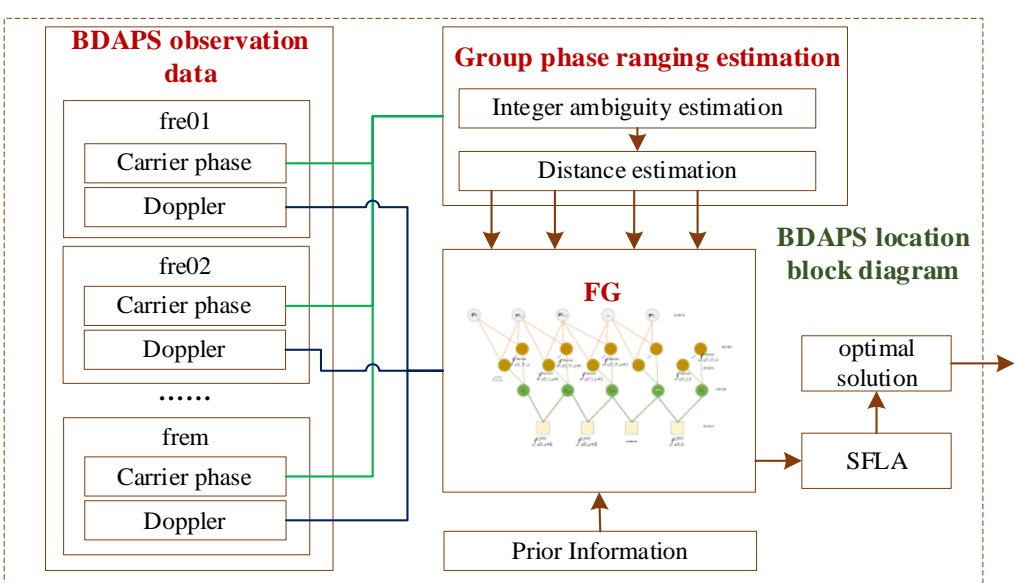

**Figure 4.** The algorithm processing block diagram.

## 4. Implementations and Evaluation

### 4.1. Real Measurement Analysis

In this section, we use the existing BeiDou pseudo-satellite test environment to analyze the ranging and positioning performance in large-scale space. In this paper, B1I (1561.098 MHz) and B2I (1207.14 MHz) are selected to broadcast the signals in the indoor

first floor hall. The location test terminal selects the time-space box location terminal which is developed by our project group based on ublox commercial chip. The data collection and display software adopts the integrated indoor and outdoor positioning APP of BeiDou. At the same time, in order to further analyze the positioning accuracy of the algorithm in this paper, the dynamic camera is selected as the location comparison benchmark. The dynamic camera can achieve real-time positioning accuracy to the millimeter level after calibration, which can meet the requirements of accuracy evaluation in this paper. The specific test scenario is shown in Figure 5. The top is the BDS pseudo-satellite positioning network, the equipment on the black shelf in the middle is the dynamic camera system, and the bottom is the positioning test service equipment that we use. The open area in the middle of the site is our actual test area. The test area has been built with accurate 3D modeling and ground test map, which can meet the needs of various test types.

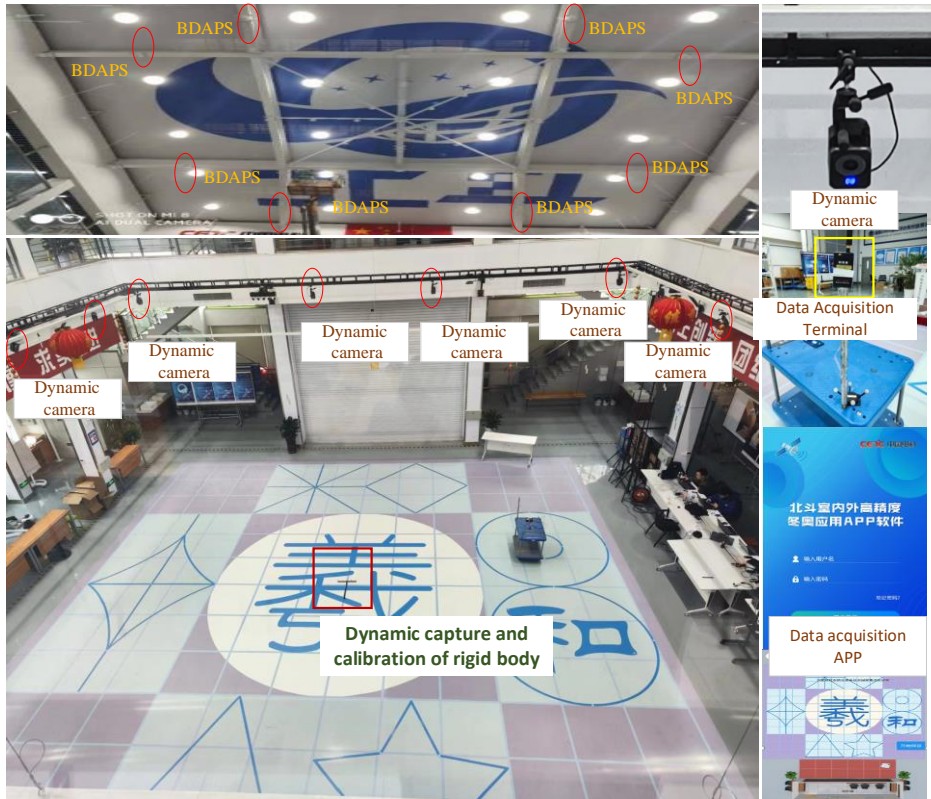

**Figure 5.** Test scenario.

Here we first broadcast the dual-frequency pseudo-satellite signal by BDAPS and validate the algorithm for the ranging capability in wireless environment. The observation information of carrier phase at 21 and 26 positions is collected for ranging analysis. As you can see from Figure 6, there is an estimation error of about one cycle in the estimation of carrier phase integer for two frequency points in the same channel, which can be up to two cycles in the previous test.

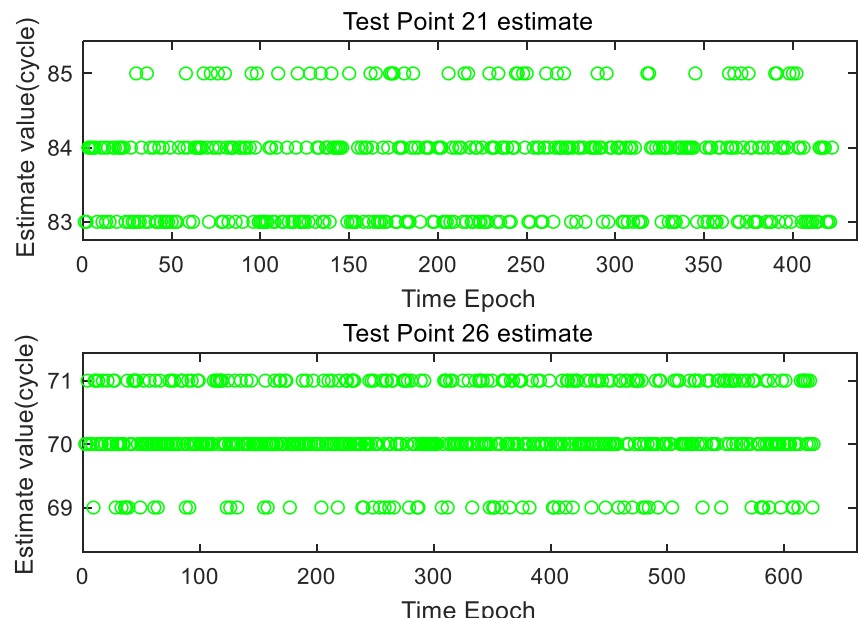

**Figure 6.** The estimation of carrier phase integer for two frequency points.

The positioning performance of the algorithm is further analyzed in this paper. During the testing process, the space-time box terminal and the dynamic capture positioning rigid body are placed near each other, and the data of the two positioning modes are collected at the same time. In order to facilitate analysis, this algorithm is represented by FG-GPRE. Here, we show the position map of two groups of motion. The detailed analysis results are as follows.

According to the test results in Figure 7, the proposed algorithm can locate continuously and does not appear the phenomenon of location hops affected by cycle slips. According to the above data, the location accuracy is further analyzed here. According to Table 3, the location ME is 17.16 cm, the location RMSE is 8.6 cm, and the maximum location error is 39.72 cm.

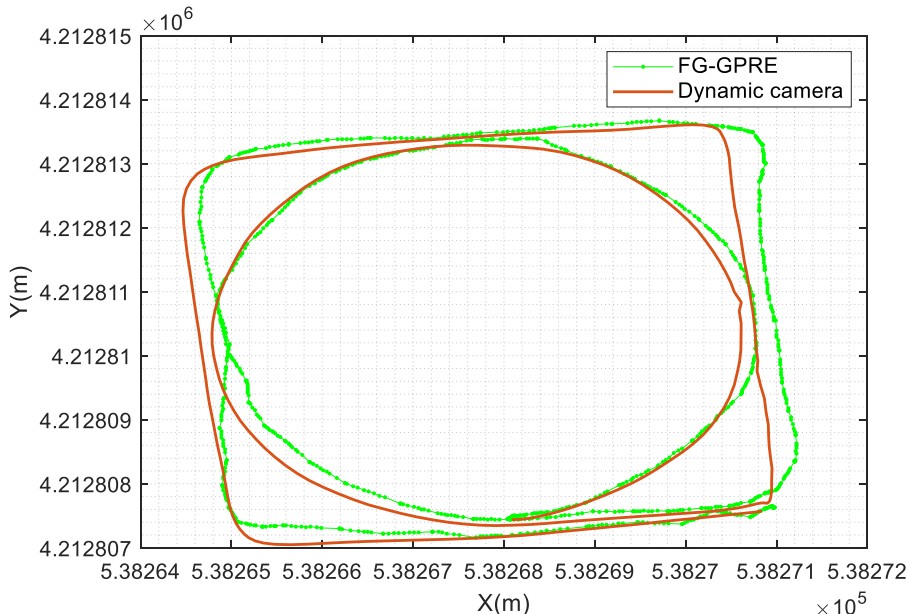

**Figure 7.** The first group of positioning accuracy comparison test.

**Table 3.** Locate the performance profiling list.

| Serial Name | ME (cm) | RMSE (cm) | MAX (cm) |
|:---:|:---:|:---:|:---:|
| FG-GPRE | 17.16 | 8.6 | 39.72 |

On the basis of the previous step, the accuracy of the positioning process is further verified here. In the same area, we have completed two rounds of dynamic positioning tests. The test results are shown in Figure 8. It is found that the two-loop trajectories obtained by the proposed algorithm are approximately coincidental and have the same positioning deviation as the reference trajectories. Although the algorithm proposed in this paper has a certain location error in the location accuracy, the location stability is improved significantly. Comparing with the literature 22, it has a better location performance. The positioning accuracy is also analyzed here. According to Table 4, the positioning ME is 20.86 cm, RMSE is 8.6 cm, and the maximum positioning error is 43.99 cm.

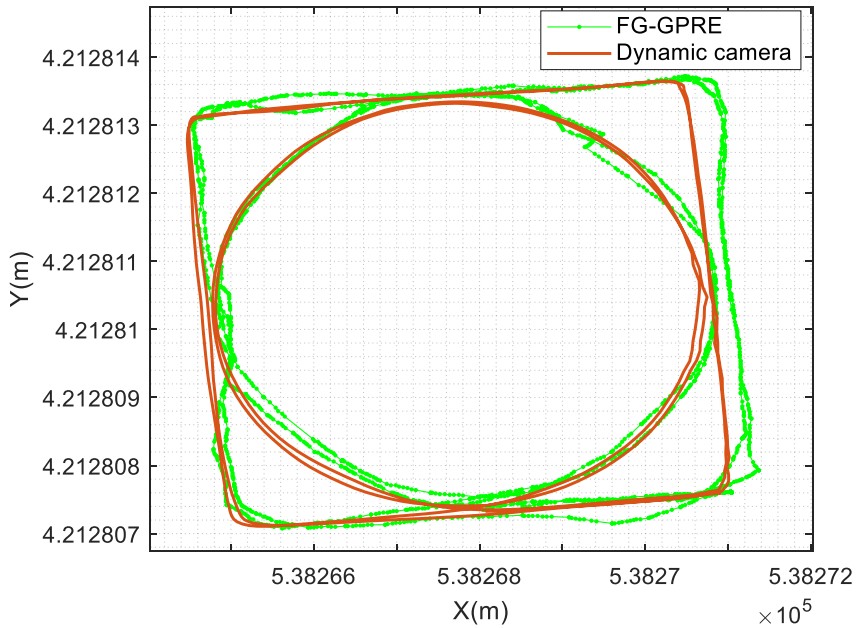

**Figure 8.** The second group of positioning accuracy comparison test.

**Table 4.** Locate the performance profiling list.

| Serial Name | ME (cm) | RMSE (cm) | MAX (cm) |
|:---:|:---:|:---:|:---:|
| FG-GPRE | 20.86 | 8.68 | 43.99 |

### 4.2. Simulation Analysis

Since the BDAPS does not have the ability to broadcast all frequency signals. In order to further verify the location performance of multi-frequency fusion ranging, the simulation experiment is carried out with the BDS data in Table 1.

A total of 24 frequency combination patterns can be found in the above analysis. In order to better verify the different group phase combination of different frequency points on the impact of ranging location, this paper selected nine of the typical combination for simulation analysis. This is shown in the Table 5.

**Table 5.** Frequency combinations in different modes.

| Name | Frequency Point Combination Mode |
|---|---|
| Mode 1 | (B1I, B1C) |
| Mode 2 | (B1I, B2a) |
| Mode 3 | (B1I, B2I/B2b) |
| Mode 4 | (B1I, B3I) |
| Mode 5 | (B1I, B1C, B2a) |
| Mode 6 | (B1C, B2a, B3I) |
| Mode 7 | (B1I, B1C, B2I/B2b) |
| Mode 8 | (B1I, B1C, B3I, B2I/B2b) |
| Mode 9 | (B1I, B1C, B2a, B3I, B2I/B2b) |

In order to ensure the maximum authenticity of the simulation data, the group phase ranging error and environmental noise are fully considered in the design of the simulation environment. Therefore, in the following analysis, carrier phase measurement data are randomly introduced with 0.02~0.04 cycles of measurement error. On this basis, the following analysis is mainly from the ranging and positioning accuracy improvement.

4.2.1. Analysis of Multi-Frequency Ranging Accuracy Improvement

The carrier phase ranging results of different frequency combinations are given in Figure 9. In the figure above, we can see from the simulation data that the characteristics of the measured values are different under different frequency combinations in the dual-frequency combination mode, the measured values under different combinations show different measurement offsets. The most obvious one is the mode l, and the measurement error is 38 cm. The mode 2 has good ranging precision, ranging error is about 10 cm. The maximum measurement error is 30 cm and the minimum measurement error is 10 cm under the mode 3. The measurement error of the mode 4 is 9 cm.

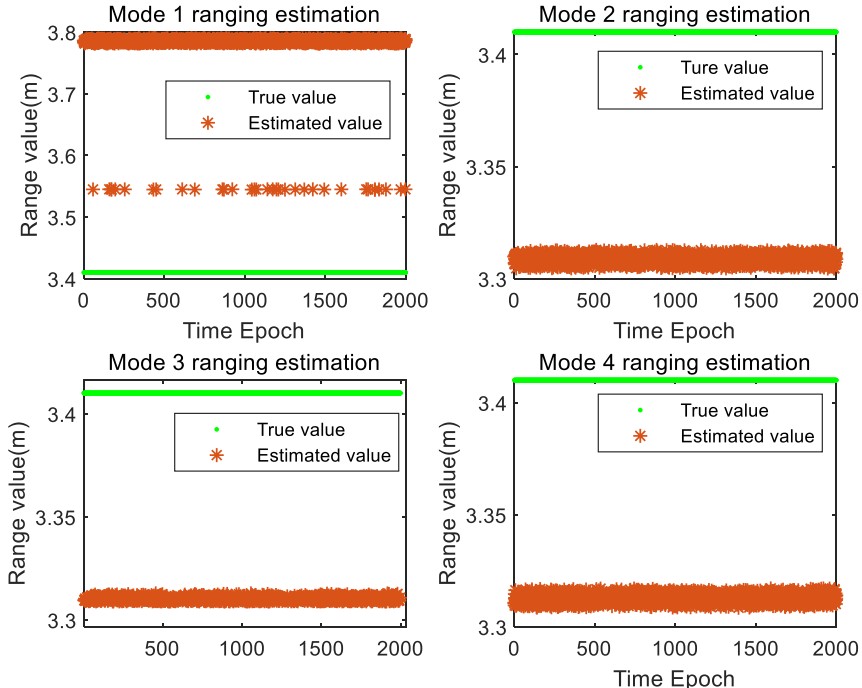

**Figure 9.** Ranging accuracy under different dual-frequency combination modes.

The Figure 10 shows the actual ranging data under three combined modes. From the above data, it is found that under three combined modes, the ranging result of mode 5 is about 3.5 m, and the ranging value of mode 6 is about 3.42 m, it can be found that the range error of the three-frequency combination mode is obviously improved compared with the

two-frequency combination, with the maximum error being about 12 cm and the minimum error being in the range of 1 cm.

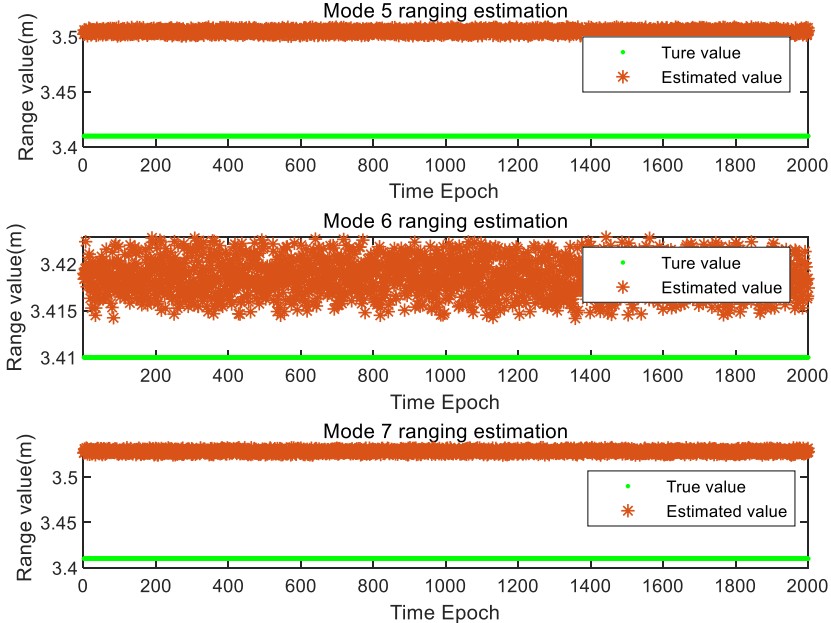

**Figure 10.** Ranging accuracy under different three-frequency combination modes.

Furthermore, Figure 11 analyzes the ranging results of four-frequency combination and five-frequency combination, and it can be found that the ranging error of the two combination modes mainly concentrates within 9 cm.

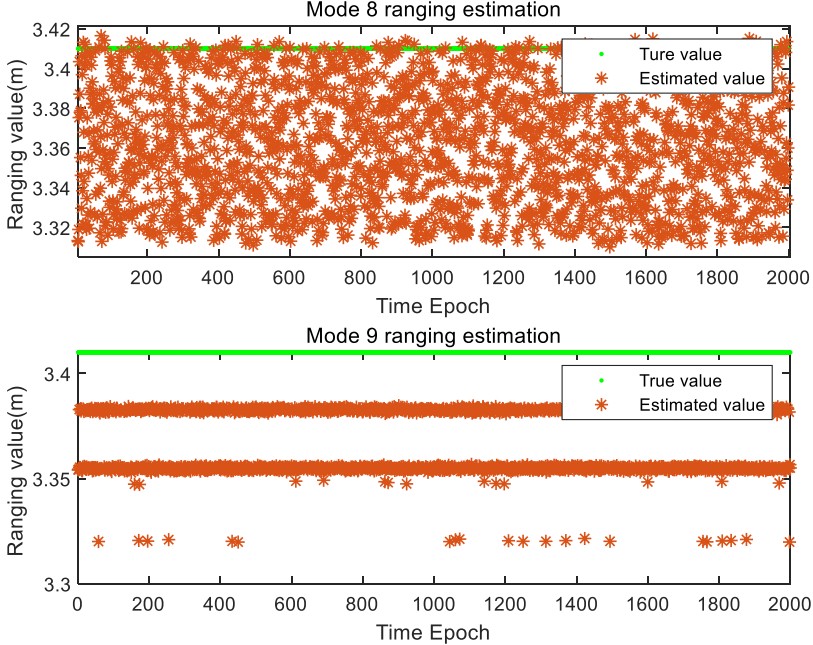

**Figure 11.** Ranging accuracy under different four-frequency combination modes.

Therefore, we further analyze the ranging mean and RMSE values under different combination modes, as shown in the following Table 6.

**Table 6.** Ranging performance under different frequency combination.

| SN | Frequency Point Combination Mode | ME (m) | RMSE | True Value (m) |
|----|----------------------------------|--------|------|----------------|
| 1 | (B1I, B1C) | 3.7814 | 0.0312 | |
| 2 | (B1I, B2a) | 3.3086 | 0.0018 | |
| 3 | (B1I, B2I/B2b) | 3.2045 | 0.0971 | |
| 4 | (B1I, B3I) | 3.3127 | 0.0018 | |
| 5 | (B1I, B1C, B2a) | 3.5048 | 0.0018 | 3.41 |
| 6 | (B1C, B2a, B3I) | 3.4186 | 0.0018 | |
| 7 | (B1I, B1C, B2I/B2b) | 3.5285 | 0.0018 | |
| 8 | (B1I, B1C, B3I, B2I/B2b) | 3.3628 | 0.0292 | |
| 9 | (B1I, B1C, B2a, B3I, B2I/B2b) | 3.3670 | 0.0147 | |

Based on the above results, the following characteristics can be found: (1) in the process of group phase ranging estimation, the larger the time-frequency difference of the multi-frequency combination mode is, the better the estimation of the carrier phase whole-cycle propagation measurement value is; (2) the introduction of more frequency points can not only improve the ranging accuracy more effectively, but also improve the whole-cycle estimation accuracy to a certain extent, three-frequency combination mode is more suitable for combination. At the same time, the combination of four-frequency and five-frequency can provide a more reliable guarantee for the scene with higher ranging accuracy and ranging stability.

### 4.2.2. Positioning Performance Analysis

According to the above results, the measurement errors will be generated randomly in the simulation process from the range errors of 30 cm, 15 cm and 10 cm. The impact on the positioning performance is further verified under different combination of range errors.

The location result is shown in Figure 12 when the range error is ±30 cm. The maximum location error in x direction is 1 m, the maximum location error in y direction is 1 m, and the maximum location error in z direction is about 0.5 m. Under the influence of space environment, the measurement error of pseudo-satellite signal is not 30 cm in any time epoch. In this case, the positioning position may deviate from the actual position, but the positioning accuracy will be better than 1 m.

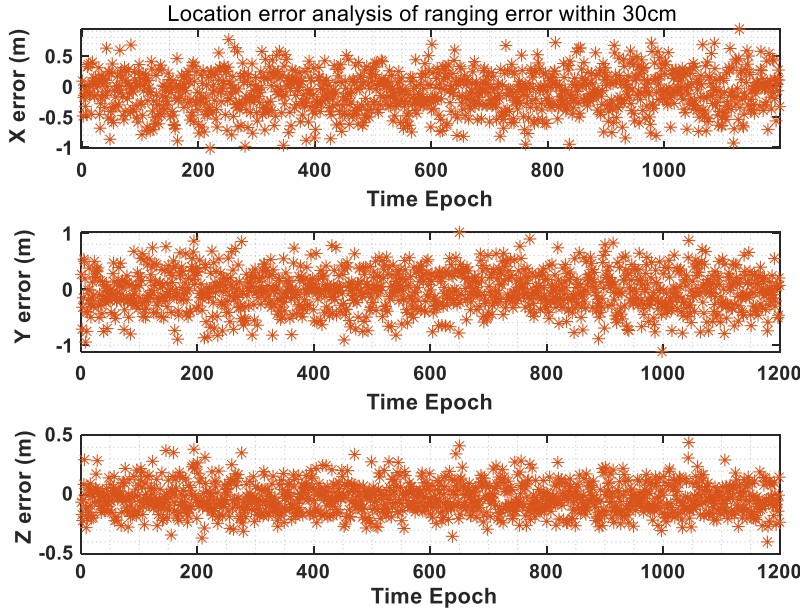

**Figure 12.** Positioning accuracy analysis when ranging accuracy is within 30 cm.

When the ranging precision is reduced to 15 cm, the location errors in three directions are obviously improved. In Figure 13, the maximum location error in x direction is 0.8 m, and the maximum location error in y direction is 1 m, the positioning error in z direction is 0.3 m. The BeiDou pseudo-satellite can achieve 0.63 m positioning accuracy in 1 confidence interval under the three-frequency combination mode.

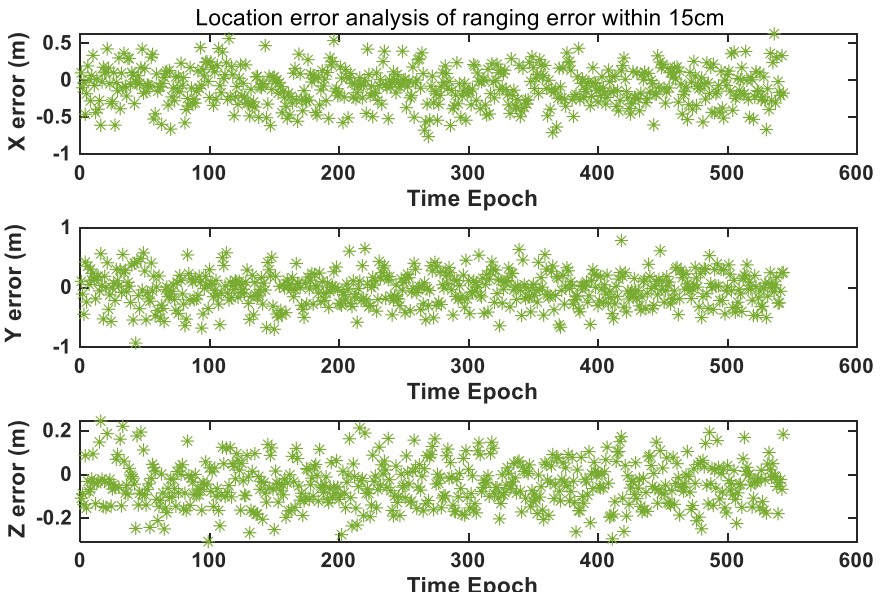

**Figure 13.** Positioning accuracy analysis when ranging accuracy is within 15 cm.

Further, it can be found that the positioning error is further significantly improved, and statistical analysis can be obtained. The position accuracy, which under 1 confidence interval, is 0.37 m. In this case, this method can well solve the problem of high-precision absolute positioning of BeiDou pseudo-satellite in indoor space (Figure 14).

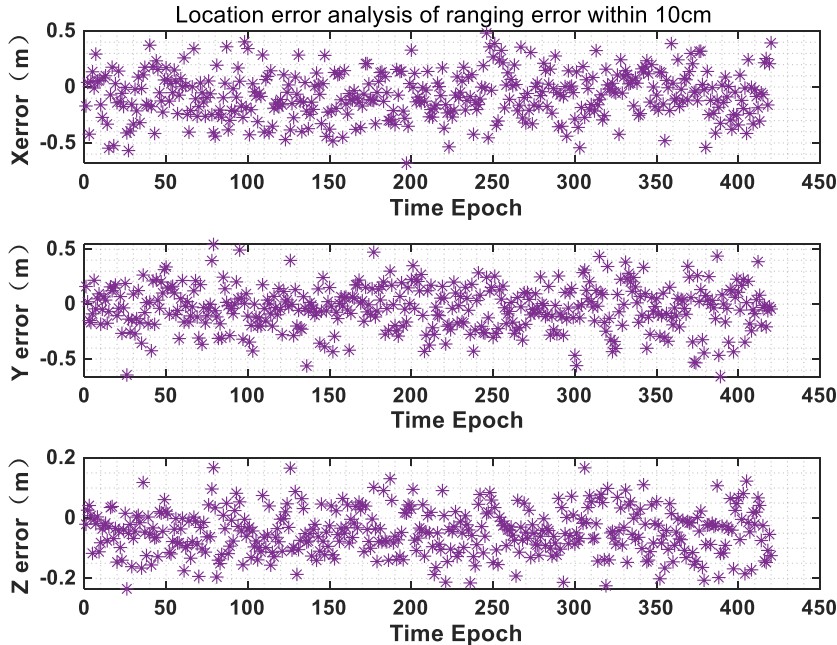

**Figure 14.** Positioning accuracy analysis when ranging accuracy is within 10 cm.

## 5. Conclusions

In the process of high precision positioning based on GNSS, the carrier phase ambiguity resolution is a necessary step. However, due to the influence of complex indoor

environments, the traditional carrier phase ambiguity estimation algorithm is difficult to achieve the effective ambiguity estimation under the influence of frequent cycle slips. The existing indoor location algorithms do not consider how to suppress the cycle slip effect on the ambiguity. This study presents a reliable indoor positioning method based on group phase ranging theory. Based on the multi-frequency phase characteristic of the BeiDou pseudo-satellite, the phase integer part of carrier phase is estimated by using the group phase theory. The high-precision indoor location is achieved by the factor map location technology assisted by SFLA. The method proposed in this paper is fully verified by the measured and simulated data. The advantages of the proposed algorithm are fully demonstrated under the condition of equal precision. In future work, we will carry out more research on the integrity of the algorithm and the reliability of location. In the existing algorithms, by fusing INS/MEMS and other multi-source sensors, a more lupin fusion localization method based on BeiDou pseudo-satellite is constructed by integrating sensors such as inertial sensors and visual radar.

**Author Contributions:** All authors together developed the idea that led to this paper. H.Z. conceived the positioning method, the experiment and the data processing of the article. S.P. provided critical comments and contributed to the final revision of the paper. All authors have read and agreed to the published version of the manuscript.

**Funding:** This research was funded by the National Key Research and Development Plan of China (project: High Precision Positioning, Navigation and Control Technology for Large Underground Space (No. s2021YFB3900800, 2021YFB3900801, 2021YFB3900802, 2021YFB3900803, 2021YFB3900804)).

**Data Availability Statement:** The data presented in this study are available on request from the corresponding author. The data are not publicly available due to ongoing projects.

**Conflicts of Interest:** The authors declare no conflict of interest.

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
