# Peer review of "LSOS: An FG Position Method Based on Group Phase Ranging Ambiguity Estimation of BeiDou Pseudolite"

_remotesensing, doi:10.3390/rs15071924_

Round 1

Reviewer 1 Report

Pseudolite is an important augmentation for indoor positioning. The authors uses a FG position method to solve the cycle slips in pseudolite positioning. The authors have presented an interesting topic. However, the whole paper should be modified significantly to tell a logical and smooth story. 

1.       The length of abstract is too long.

2.       It is strange to use lots of equations in the introduction part. Please consider changing these equations to statements, and try to cite more papers.

3.       Abbreviations must be defined clearly upon first use. For example, FG in line 22, TOA in Line 41, and GNSS in line 40.

Reviewer 2 Report

In this paper, based on the principle of group phase period quantization, the multi-frequency characteristic of Beidou pseudo-satellite is used to estimate the carrier phase propagation ambiguity, and parameter estimation using factor graph. Compared with other methods, the experimental results have some improvement and are of reference value.

Question 1

Ensure that the first occurrence of keyword abbreviations throughout the text carries a specific meaning, e.g. factor graph (FG) in the abstract.

Question 2

Please simplify the full description of the equation, e.g. equation (1) to (4) can be expressed directly as equation (4).

Question 3

Which table does "Table XXX" in line 546 of the text specifically refer to?

Question 4

Please re-describe the “Conclusions” and “Discussion” separately. Section 5 in the current paper is conclusions, not a discussion.

Reviewer 3 Report

Dear author,

thank you very much  for your contribution “LSOS: An FG Position Method based on Group Phase Ranging Ambiguity Estimation of Beidou Pseudolite”. In this article, you analysed the usage of pseudolite of the Beidou satellite system for indoor navigation. You use the factor graph technology for optimization in non-linear problems.

I think your approach is very interesting. Nevertheless, your article is very long. At the beginning and at the end you can skip many plots and maybe make plots, which presents the differences between the different modes much more to the point.

So on page 2 you can get very quickly from the phase measurement equation 1 to the phase differences in equation 5 and skip the equations in between.

Figures 1 and 2 can also be skipped, as they only should show the occurrence of cycle slips which you are 0.5 times per second.

Figure 3 only shows a schema of the pseudolite, not interesting for the reader.

Figure 4: If you skip the third frequency and make the phase difference more visible. The point you want to show would be much more prominent.

Equation 6 can be skipped.

Equation 14 skipped and equation 13 and 15 as one equation.

Simulation can be explained in more detail. Is it a static simulation? Did you introduce cycle slips?

Nearly all plots miss the units of the axes.

Figure 11 must be more precise: What is the true range? What causes the difference in noise and bias and where do the outliers come from? Is one plot for the modes with the same number of frequencies enough? Is it better to show the difference to the true range? Mode 2 is in there double.

You miss a conclusion. How would you proceed with this method? What can be done better?

In general your English can be understood, but especially in the beginning it need to be improved: You make long sentences with incorrect grammar. I found it very difficult to read and understand what you want to say.

Example: Line 19 to 22 in the abstract.

Lines 76 to82.

Line 123-126

Lines 225 – 239

To better understand your optimisation: can you compare your method with a Kalman Filter?

Some typos:

Line 368 Beidou

Line 97 Doppler

Best regards.

Round 2

Reviewer 3 Report

Dear autor,

thank you very much for taking my comments into account. I think your paper improved a lot.

Best regards.